# Relation between VT1, VT2, and VO_2max_ with the Special Wrestling Fitness Test in Youth Wrestlers: A Short Report

**DOI:** 10.3390/ijerph20032570

**Published:** 2023-01-31

**Authors:** Tomás Herrera-Valenzuela, Emerson Franchini, Pablo Valdés-Badilla, Alex Ojeda-Aravena, Carolina Pardo-Tamayo, Carolina Zapata-Huenullán, Cristián Cofre-Bolados, Celso Sanchez-Ramirez

**Affiliations:** 1Facultad de Ciencias Médicas, Escuela de Ciencias de la Actividad Física, el Deporte y la Salud, Universidad de Santiago de Chile (USACH), Santiago 9170022, Chile; 2Sports Department, School of Physical Education and Sport, University of São Paulo, São Paulo 05594-110, Brazil; 3Department of Physical Activity Sciences, Faculty of Education Sciences, Universidad Católica del Maule, Talca 3530000, Chile; 4Sports Coach Career, School of Education, Universidad Viña del Mar, Viña del Mar 2520000, Chile; 5IRyS Group, Physical Education School, Pontificia Universidad Católica de Valparaíso, Valparaíso 2581967, Chile; 6Escuela de Ciencias del Deporte, Facultad de Salud, Universidad Santo Tomás, Santiago 8370003, Chile

**Keywords:** combat sports, athletic performance, physical conditioning

## Abstract

This study investigated the relationship between peak oxygen uptake and ventilatory threshold 1 (VT1) and 2 (VT2) with the Special Wrestling Fitness Test variables. Thirteen wrestlers (male: six; female: seven) of Olympic freestyle wrestling were assessed. The Pearson’s correlation coefficient (*p* < 0.05) was used to establish the relationship between variables. A positive correlation was found between VT1 with throws in set B (r = 0.77; *p* = 0.002; 95%CI = 0.37–0.93), total throws (r = 0.73; *p* = 0.004; 95%CI = 0.30–0.91), heart rate recovery (r = 0.58; *p* = 0.036; 95%CI = 0.05–0.86), and test index (r = −0.60; *p* = 0.031; 95%CI = −0.86–0.07); between VT2 and throws in set B (r = 0.57; *p* = 0.043; 95%CI = 0.01–0.86); and between peak oxygen uptake with throws in set B (r = 0.77; *p* = 0.002; 95%CI = 0.39–0.93), throws in set C (r = 0.64; *p* = 0.02; 95%CI = 0.12–0.89), and total throws (r = 0.72; *p* = 0.006; 95%CI = 0.28–0.91). In conclusion, the peak oxygen uptake and ventilatory thresholds correlated with specific Special Wrestling Fitness Test variables.

## 1. Introduction

A wrestlers performance is influenced by their cardiorespiratory fitness [1]. In particular, the aerobic system contributes to maintaining the effort throughout the combats and stimulating the recovery process between periods [1]. Thus, aerobic power, assessed via maximum oxygen uptake (VO_2max_)-graded exercise tests, is one of the most popular physiological variables in this sport [1]. Values that vary between 37 mL/kg/min and 67 mL/kg/min for males and between 39 mL/kg/min and 52 mL/kg/min for females have been reported [1]. However, the cardiorespiratory fitness of wrestlers is usually assessed through general tests using a treadmill or cycle ergometer [1] that do not represent the specific characteristics of the sport. Although wrestling-specific tests have been developed, currently there is no specific test that determines the VO_2max_ of wrestlers [2]. Nonetheless, it has recently been established that the Special Wrestling Fitness Test (SWFT) is related to aerobic performance [3,4,5].

The SWFT presents different variables, such as the number of throws, the heart rate immediately after the test (HR_final_), the heart rate one minute after the test (HR_1min_), and the test index (SWFT_index_) calculated as HR_sum_/throws [3,4,5,6,7]. However, recent studies have only found a positive significant correlation between the VO_2max_ with the number of throws (r = 0.821 to 0.829) [5] and consequently with the SWFT_index_ (r = −0.325 to −0.815) [3,5], without finding a significant correlation between the VO_2max_ and HR_final_, HR_1min_, and HR_sum_ [3,4,5]. On the other hand, the VO_2max_ during SWFT has been measured directly and a significant positive correlation (r = 0.93) was reported with the VO_2max_ measured during an incremental treadmill test [3]. Still, to our knowledge, other submaximal variables of aerobic performance, such as ventilatory threshold 1 (VT1) and ventilatory threshold 2 (VT2), have not been measured in wrestlers. Their relationship with SWFT performance is unknown, which may be relevant since VT1 and VT2 are related to performance during interval effort in team sports [8]. Indeed, previous studies with judo athletes have investigated the association between physical fitness and SJFT [9,10,11], reporting a significant positive correlation between the anaerobic velocity threshold measured in a treadmill test and the number of throws in Special Judo Fitness Test (SJFT) (r = 0.60) [9]. 

Knowing the relationship of ventilatory thresholds with SWFT performance may provide new information on the criterion validity of the SWFT [12]; in addition, it could be helpful for coaches to provide more details about a specific field test that is simple to apply and inexpensive in terms of technology use. 

Therefore, the objective of the present investigation was to investigate the relationship between the VO_2max_, VT1, and VT2 with the SWFT variables (HR_final_, HR_1min_, HR_sum_, throws, and SWFT_index_). Based on previous studies [9,10], we hypothesize that the VO_2max_, VT1, and VT2 will correlate significantly with SWFT variables.

## 2. Materials and Methods

### 2.1. Participants

A sample was selected for the convenience of 13 wrestlers belonging to the Chilean national team, distributed in 6 male freestyle (Freestyle) (age: 16.0 ± 1.4 years; height: 1.71 ± 0.01 m; body mass: 69.5 ± 14.9 kg; age category: 5 cadets and 1 junior; weight division: −51 kg, −60 kg, −65 kg, −74 kg, −80 kg, and −92 kg) and 7 female athletes (Women’s Wrestling) (age: 13.7 ± 0.8 years; height: 1.61 ± 0.04 m; body mass: 59.7 ± 2.9 kg; age category: 7 cadets; weight division: −46 kg, −53 kg, −57 kg, −61 kg, −65 kg, −69 kg; and −73 kg), were assessed.

The inclusion criteria were: (a) at least two years of experience in wrestling practice; (b) participate in at least five training sessions per week; (c) be in a competitive period; and (d) have at least two months of uninterrupted training. The exclusion criteria were: (a) having an injury or physical disorder that would remove them from practicing sports; (b) consuming any nutritional supplement or medication affecting performance; and (c) being in the process of rapid weight loss. All the athletes have experience in international competitions and obtained medals during the South American Wrestling Championship, Santiago, 2019.

All the participants were informed verbally and in writing about the study’s purpose, methods, and means. The athletes had previous experience in the test, although they received a feedback session for correct execution and to avoid the learning effect. The participants’ parents signed a consent authorizing the use of the information for scientific purposes, while the athletes were also asked to sign the support. The research protocol was reviewed and approved by the Scientific Ethics Committee of the Universidad Santo Tomás de Chile (Code: 43.18) and was developed following the Declaration of Helsinki.

### 2.2. Procedures and Measures

The measurements were carried out at the Olympic Training Center in Chile in November 2021. To execute the incremental treadmill test, the participants used a CORTEX Metamax^®^ 3B portable gas analyzer (Cortex Biophysik GmbH Leipzig, Germany) and a Polar heart rate monitor model H10 (Polar Inc., Kempele, Finland).

The participants reported to the laboratory for two non-consecutive sessions, separated by 72 h, performed the SWFT measurement and an incremental running test. The participants arrived at the laboratory at 8:00 a.m. Before each measurement, they were asked to refrain from exercising beyond what was required for the study and to maintain their regular diet.

Special Wrestling Fitness Test (SWFT): Before the test, the participants completed a 20 min warm-up, which included general and specific wrestling exercises that athletes normally perform during training. The participants were familiarized with the test and the material before each evaluation to avoid any learning effect that could explain the improvement in actions over time. The test was performed on a wrestling mat (Dollamur, Fort Worth, Texas, USA) approved by United World Wrestling for international competitions, with the athlete throwing two other wrestlers of the same weight division and similar height (who were 6 m apart from each other) as many times as possible in three sets of 15 s, 30 s, and 30 s, respectively, with 10 s of rest between them [3,4,5,6,7] (see Figure 1). As in previous studies, the Freestyle and Women’s wrestlers used the fireman’s carry technique [3,5]. This procedure aimed to determine the heart rate values immediately after the test (HR_final_), heart rate immediately after 1 min of recovery (HR_1min_), and the total number of throws. The SWFT_index_ was calculated using the following equation: SWFT_index_ = (HR_final_ + HR_1min_)/throws). 

### 2.3. Incremental Test

The incremental test began at 6.4 km/h, with the speed increasing by 1.6 km/h (1 mile/h) each minute and ending with the participant’s voluntary exhaustion. Throughout the test, the treadmill’s incline was kept at 1%. The test was considered maximal when the participants met two of the following criteria: (1) HR > 95% of the maximum theoretical HR; (2) VCO_2_/VO_2_ > 1.1; (3) rating of perceived exertion from 19 to 20 (RPE 6–20); and (4) VO_2_ plateau. To determine the VT1 and VT2, the graphs proposed by Wasserman, numbers six and nine (ventilatory equivalents and final expiratory pressure of oxygen and carbon dioxide, respectively), were used. The VT1 was found when an increase in the VE/VO_2_ and end-tidal PO_2_ (PETO_2_) without a concomitant increase in VE/VCO_2_ was observed. At the same time, the VT2 was determined when an increase in the VE/VO_2_ and VE/VCO_2_ and a decrease in end-tidal PCO_2_ (PETCO_2_) were observed [13].

### 2.4. Statistic Analysis

The statistical program SPSS^®^ version 26.0 was used for the analysis. The data are presented as mean and standard deviations with their respective 95% confidence interval. The analyzed outcomes complied with the normality of data through the Shapiro–Wilk test. The Pearson product moment correlation test for parametric variables was used to establish the relationship between the variables. The correlation magnitudes were interpreted following thresholds from 0 to 0.30 (low); from 0.31 to 0.49 (moderate); from 0.50 to 0.69 (large); from 0.70 to 0.89 (very large); and from 0.90 to 1.0 (a near perfect to perfect correlation). The level of statistical significance was set at *p* < 0.05. 

## 3. Results

Table 1 presents the physiological response during the Special Wrestling Fitness Test (HR_final_, HR_1min_, HR_sum_, throws, and SWFT_index_) and the incremental test (VT1, VT2, and VO_2max_). 

Figure 2 presents the correlation between physiological variables of the incremental test and the Special Wrestling Fitness Test. Panels A, B, and C show significant large correlations in the number of throws in series b with the VT1, VT2, and VO_2max_. Panel E reports a large correlation between the total number of throws in the SWFT with the VT1. Panel D and F present significant correlations of throws set c and total throws with the VO_2max_. Figure G and H illustrate moderate to large correlations of the VT1 with HR_1min_ and SWFT_index_. No significant correlation was found for the VT1 variable with HR_final_ (r = −0.10, *p* = 0.746, CI = from −62 to 0.48), HR_sum_ (r = 0.39, *p* = 0.192, CI = from −0.21 to 0.77), and throws in set A (r = 0.48, *p* = 0.101, CI = from −0.10 to 0.81); for the VT2 variable with HR_final_ (r = −0.49, *p* = 0.095, CI = from −0.82 to 0.11), HR_1min_ (r = 0.26, *p* = 0.384, CI = from −0.36 to 0.72), HR_sum_ (r = 0.09, *p* = 0.760, CI = from −0.50 to 0.62), throws in set A (r = 0.31, *p* = 0.298, CI = from −0.31 to 0.75), throws in set C (r = 0.54, *p* = 0.061, CI = from −0.04 to 0.85), total throws (r = 0.52, *p* = 0.074, CI = from −0.07 to 0.84), and the SWFT_index_ (r = −0.49, *p* = 0.092, CI = from −0.83 to 0.10); and for the VO_2max_ variable with HR_final_ (r = −0.26, *p* = 0.395, CI = from −0.71 to 0.34), HR_1min_ (r = 0.31, *p* = 0.305, CI = from −0.29 to 0.74), HR_sum_ (r = 0.12, *p* = 0.702, CI = from −0.46 to 0.63), and throws in set A (r = 0.39, *p* = 0.186, CI = from −0.20 to 0.78).

## 4. Discussion

The aim of this study was to relate VO_2max_, VT1, and VT2 with SWFT variables (HR_final_, HR_1min_, HR_sum_, throws, and SWFT_index_). We hypothesized that VO_2max_, VT1, and VT2 would correlate with all the SWFT variables. In this sense, the hypothesis was fulfilled. The main findings of the present study were significant positive correlations between the VO_2max_ with the number of B-set throws, C-set throws, and the negative correlation between total throws and SWFT_index_. Between the VT1 with the number of B-set throws, total throws, HR_1min_, and the SWFT_index_, and between the VT2 with the number of B-set throws.

Our results suggest that the different SWFT variables are associated with specific aerobic performance markers and are consistent with recent studies that have reported a significant correlation between the VO_2max_ and SWFT [3,4,5]. The positive correlation between the VO_2max_ and the SWFT is likely due to the fact that a higher VO_2max_ is associated with a faster phosphocreatine resynthesis. A higher phosphocreatine store is important for repeated high-intensity actions interspersed by short recovery durations [14], which are the characteristics of the SWFT. Indeed, for the SJFT it was reported that there was a higher participation of the phosphagen system for the total energy expenditure [15]. A new result found in our study was the correlation between the directly measured VO_2max_ and the number of throws in sets B and C. However, the VO_2max_ was not correlated with set A, probably due to low oxidative participation in the first 15 s of this test [15] and due to the increase in the VO_2_ over time during SWFT [3].

Furthermore, to our knowledge, this is the first study to present the correlation between the VT1 and VT2 with the SWFT variables. In this context, we found that the VT1 and VO_2max_ correlated significantly with the number of throws in the B set, the number of total throws, and the SWFT_index_. Using another submaximal variable, corresponding to the intensity of the second threshold, Detanico et al. [9] reported a similar correlation between the anaerobic threshold velocity and the number of throws during the SJFT [9]. Our results also agree with a recent study that assessed judo athletes and untrained subjects [16], where a correlation was found between VT1 with a decrease in the peak power output and accumulated work for the upper body during the repeated sprint ability (RSA). However, when the judo athletes were analyzed separately, the VT1 was not correlated with the RSA [16]. In addition, the VT1 is the only variable correlated with recovery capacity at the end of the test, expressed through HR_1min_. 

The exercise above the ventilatory threshold is associated with a non-linear increase in lactate values and the level of fatigue. The ventilatory threshold values usually are between 50% and 80% of the VO_2max_, even in trained athletes, while wrestlers must compete at high-intensity efforts [17]. However, the effort during the wrestling competition is intermittent [18,19] and having high values of the ventilatory threshold could improve recovery for the next high-intensity effort.

However, the VT2 only correlated with the number of throws in set B. This finding disagrees with previous studies, where OBLA, corresponding to the second threshold, was associated with the RSA in soccer players [20] and the VT2 was correlated with RSA in ice hockey athletes [8].

One limitation of this study is the small sample of male and female athletes. However, this is normal when investigating high-level athletes since accessing large samples of athletes with these characteristics is complex. On the other hand, this is the first study that analyzed the relationship between SWFT with different maximal and submaximal aerobic fitness variables (VT1, VT2, and VO_2max_), providing new insights into SWFT’s criterion validity [12]. However, these results are not generalized to senior athletes. Future studies could use a more significant number of male and female athletes separately, verify the results by test–retest, and analyze other components of the SWFT performance, such as its relationship with neuromuscular performance. 

On the other hand, although the contribution of energy systems during wrestling competitions has yet to be discovered, some studies through time motion analysis show that the aerobic contribution could reach up to 79% [21,22]. However, to estimate the contribution of energy systems more clearly, it is necessary to measure physiological variables such as lactate and oxygen uptake. Therefore, studies with these characteristics have been carried out with karate, boxing, judo, and taekwondo athletes, finding a contribution aerobic of 70% [23], 77% [24], 70% [25], and 66% [26], respectively. In any case, the aerobic system plays a key role during wrestlers’ competitions. Therefore, coaches plan aerobic training in search of specific adaptations, such as VT1, VT2, and VO_2_; however, no wrestling test allows for estimating these variables [2]. Therefore, our study enables coaches to use the different variables of the SWFT to control aerobic adaptation.

## 5. Conclusions

In conclusion, this study reports that the volume oxygen uptake, ventilatory threshold 1, and ventilatory threshold 2 are correlated with the specific SWFT variables. Therefore, in practical terms, our results can be used by wrestling coaches who seek to control the aerobic fitness of their athletes through a specific test that is easy to apply.

## Figures and Tables

**Figure 1 ijerph-20-02570-f001:**
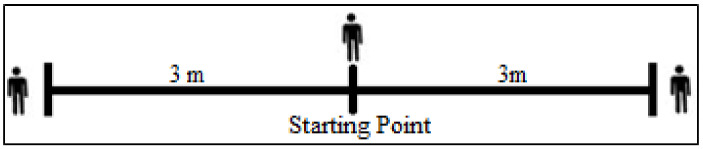
Special Wrestling Fitness Test.

**Figure 2 ijerph-20-02570-f002:**
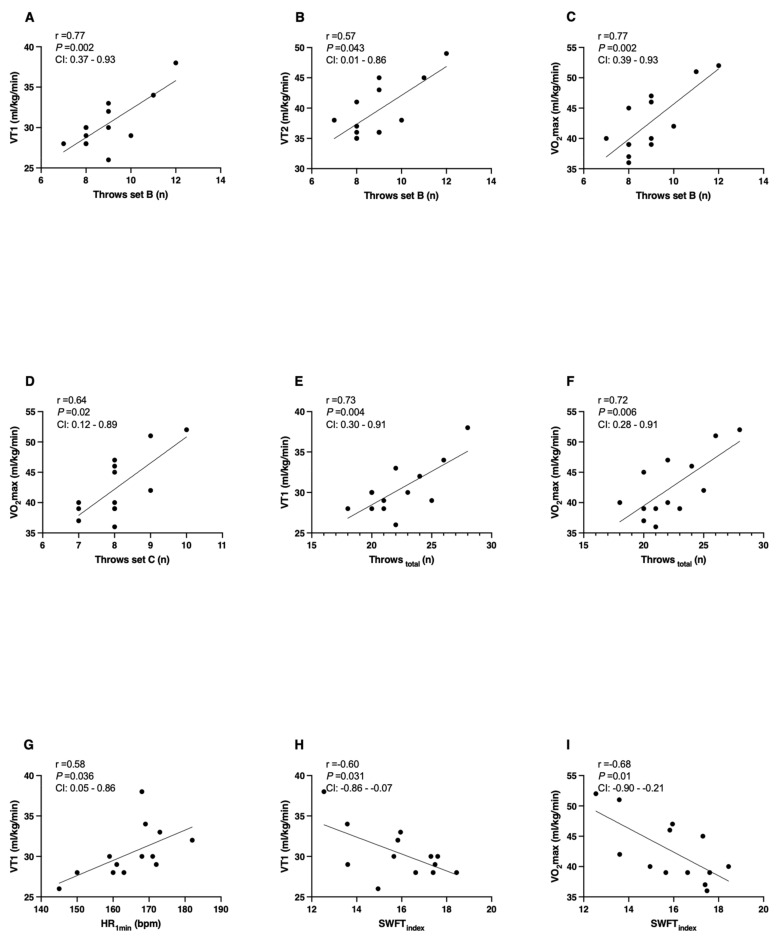
Correlation between physiological variables of the incremental test and the Special Wrestling Fitness Test. (Panel (**A**)): correlation between ventilatory threshold 1 and the number of throws during set B; Panel (**B**)): correlation between ventilatory threshold 2 and the number of throws during set B; (Panel (**C**)): correlation between maximum oxygen uptake and the number of throws during set B; (Panel (**D**)): correlation between maximum oxygen uptake and the number of throws during set C; (Panel (**E**)): correlation between ventilatory threshold 1 and the number of total throws; (Panel (**F**)): correlation between the maximum oxygen uptake and the number of total throws; (Panel (**G**)): correlation between ventilatory threshold 1 and heart rate after 1 min of recovery; (Panel (**H**)): correlation between ventilatory threshold 1 and the index of the Special Wrestling Fitness Test; and (Panel (**I**)): correlation between maximum oxygen uptake and the number of total throws.

**Table 1 ijerph-20-02570-t001:** Physiological response during the Special Wrestling Fitness Test and the incremental test in national-level wrestlers.

Variable	Mean ± SD	CI 95%
**Special Wrestling Fitness Test**		
HR_final_ (bpm)	186 ± 6	183–189
HR_1min_ (bpm)	165 ± 10	159–171
HR_sum_ (bpm)	351 ± 14	343–359
Throws in set A (n)	5 ± 1	5–6
Throws in set B (n)	9 ± 1	8–10
Throws in set C (n)	8 ± 1	8–9
Throws_total_ (n)	22 ± 3	21–24
SWFT_index_ (A.U.)	15.92 ± 1.81	14.82–17.01
**Incremental Test**		
VT1 (mL/kg/min)	30.38 ± 3.18	28.47–32.30
VT2 (mL/kg/min)	39.54 ± 4.59	36.76–42.31
VO_2max_ (mL/kg/min)	42.54 ± 5.19	39.40–45.67

HR_final_: Heart Rate final on Special Wrestling Fitness Test; HR_1min_: Heart Rate 1 min recovery on Special Wrestling Fitness Test; HR_sum_: Heart Rate sum on Special Wrestling Fitness Test; Throws: number of throws on Special Wrestling Fitness Test index; SWFT_index_: Special Wrestling Fitness Test index; VT1: Ventilatory Threshold 1; VT2: Ventilatory Threshold 2; and VO_2max_: maximum oxygen uptake.

## Data Availability

The datasets generated during and/or analyzed during the current research are available from the Corresponding author on reasonable request.

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
