# Peer review of "Relation between VT1, VT2, and VO2max with the Special Wrestling Fitness Test in Youth Wrestlers: A Short Report"

_ijerph, 2023, doi:10.3390/ijerph20032570_

Round 1

Reviewer 1 Report

The study aimed to investigate the relationship between peak oxygen consumption and ventilatory threshold 1 and 2 with the Special Wrestling Fitness Test variables. It is an interesting and well-conducted study, I have some minor questions and suggestions.

- I believe the manuscript provides an important information on the criterion validity of the SWFT. Therefore, I encourage the authors to include this information as additional justification (introduction) and in the practical applications (end of discussion). For additional information, see previous studies (DOI: 10.2165/00007256-200838040-00003, DOI: 10.3389/fphys.2018.00386) 

- Please, add more information on the athletes' competitive level

- Were the correlations between SWFT set A and aerobic variables carried out? There is a mention in the discussion, but this information should be clear in the results, providing the r and p values, even if not signficant.

- The literature in general is confused regarding the different nomenclatures of aerobic thresholds. It is important to make it clear which index is being compared in the discussion. For example, anaerobic threshold velocity (Detanico et al.) corresponds to VT2 not VT1; OBLA is referente to VT2.

Reviewer 2 Report

Dear Authors

You have written an interesting paper. However, some parts need to be addressed for greater clarity.

The title should mention the youth population (in youth wrestlers) so it is clear these are not senior athletes. Please amend

The introduction is on point and clearly leads to the main study rationale.

Line 66 - a recent study connected to the SJFT test could be helpful here and in later discussion as it discussed the performance tests connected to the SJFT (https://www.mdpi.com/2411-5142/7/4/101).

Methods: How was your sample size determined (G*Power or any other method) Please report

Please report participants' competition weight categories.

Be specific - Chilean junior or cadet national team (or perhaps even senor). Report

Line 97 / please report the correct manufacturer information (Polar,.... Finland,. etc)

Line 109 / what was the weight of those 2 partners? were they the same weight and height? Report

Limitations should mention the youth of athletes and that these findings should not be generalised for senor athletes and separate studies should be made on a greater sample of male and female athletes separately.

The conclusion paragraph should be extended with some more pointers for the practical applicability of this test for wrestling coaches and athletes. Please amend (Try to move the last sentence of the discussion in the conclusion)

Overall the paper is solid, however, still needs some more work.

Kind regards

Round 2

Reviewer 2 Report

Dear Authors,

Thank you for addressing all raised questions adequately. The manuscript quality has improved. 

Therefore, I recommend acceptance in the present form

Kind regards and congratulations to the authors.